# Hypokalemia is frequent and has prognostic implications in stable patients attending the emergency department

Laure Abensur Vuillaume[1,2,3], João Pedro Ferreira[1,3], Nathalie Asseray[4], Béatrice Trombert-Paviot[5], Emmanuel Montassier[3,6], Matthieu Legrand[7], Nicolas Girerd[1,3], Jean-Marc Boivin[1,3], Tahar Chouihed[1,3,8], Patrick Rossignol[1,3]*

1 Université de Lorraine, Inserm, Centre d'Investigations Cliniques- Plurithématique 1433, and Inserm U1116, CHRU, Nancy, France, 2 Emergency Departement, Regional Hospital Metz-Thionville, Metz, France, 3 F-CRIN INI-CRCT (Cardiovascular and Renal Clinical Trialists), France, 4 Infectious Diseases Department, Nantes University Hospital and CIC 1413, INSERM, Nantes, France, 5 Department of Public Health and Medical Informatics, University Hospital of Saint Etienne and Host Research Team SNA-EPIS, PRES Lyon, Jean Monnet University, Lyon, France, 6 Department of Emergency Medicine, Nantes University Hospital; MiHAR lab, Université de Nantes, Nantes, France, 7 APHP, Department of Anaesthesiology, Critical Care Medicine and Burn Unit, Saint Louis University Hospital, INSERM UMR-S942, INI-CRCT network and Univ Paris Diderot, Paris, France, 8 Emergency Departement, University Regional Hospital, Nancy, France

* p.rossignol@chru-nancy.fr

## Abstract

### Background

Potassium disturbances are associated with adverse prognosis in patients with chronic conditions. Its prognostic implications in stable patients attending the emergency department (ED) is poorly described.

### Aims

This study aimed to assess the prevalence of dyskalemia, describe its predisposing factors and prognostic associations in a population presenting the ED without unstable medical illness.

### Methods

Post-hoc analysis of a prospective, cross-sectional, multicenter study in the ED of 11 French academic hospitals over a period of 8 weeks. All adults presenting to the ED during this period were included, except instances of self-drug poisoning, inability to complete self-medication questionnaire, presence of an unstable medical illness and decline to participate in the study. All-cause hospitalization or deaths were assessed.

### Results

A total of 1242 patients were included. The mean age was 57.2±22.3 years, 51% were female. The distribution according to potassium concentrations was: hypokalemia<4mmol/L (n = 620, 49.9%), normokalemia 4-5mmol/L(n = 549, 44.2%) and hyperkalemia >5mmol/L(n

**Data Availability Statement:** There are ethical restrictions for data access per French Regulation (sensitive healthcare data). Requests for access

require the agreement of the study investigators and the study sponsor studying the applicant's project. The contact person for this committee is Ms Béatrice Trombert-Paviot. (Beatrice. TROMBERT-PAVIOT@chu-st-etienne.fr; CHU of Saint Etienne, France) or Pr Patrice Queneau (pat-queneau@orange.fr, Saint Etienne University, France).

**Funding:** The National Medicine Academy approved the study design and provided funding. The French Society for Clinical Pharmacy (SFPC) approved the protocol, supported the enrolment and training of pharmacy students and provided funding. The French Association of Pharmaceutical Manufacturers for a responsible self-medication (AFIPA) approved the protocol and the conduct of the study. Sanofi Aventis approved the protocol and the conduct of the study. Patrick Rossignol reports receiving consulting fees and travel support from Novartis, consulting fees from Novo Nordisk, AstraZeneca, Grünenthal, and Corvidia, consulting fees, lecture fees, fees for serving on a steering committee, and travel support from Relypsa/Vifor/ Vifor Fresenius Medical Care, fees for serving on a steering committee and fees for serving on a critical event committee from Idorsia, lecture fees and travel support from Bayer and Servier, owning stock options in G3 Pharmaceuticals, and fees for serving as co-founder and owning stock in CardioRenal. TC reports honoraria from Novartis, Astra Zeneca. ML reports receiving lecture fees from Baxter and Fresenius, research support from Sphingotec, and consulting fees from Novartis. None of these funding sources intervened in the collection, management, analysis or interpretation of the data; nor in the preparation, review or approval of the manuscript.

**Competing interests:** Importantly, there is no funding source for this post-hoc analysis of the ADES-ED cohort study, while the authors of the princeps manuscript "have declared no potential conflicts of interest regarding this study." For the present manuscript: Patrick Rossignol reports receiving consulting fees and travel support from Novartis, consulting fees from Novo Nordisk, AstraZeneca, Grünenthal, and Corvidia, consulting fees, lecture fees, fees for serving on a steering committee, and travel support from Relypsa/Vifor/ Vifor Fresenius Medical Care, fees for serving on a steering committee and fees for serving on a critical event committee from Idorsia, lecture fees and travel support from Bayer and Servier, owning stock options in G3 Pharmaceuticals, and fees for serving as co-founder and owning stock in CardioRenal. TC reports honoraria from Novartis, Astra Zeneca. ML reports receiving lecture fees

= 73, 0,6%). The proportion of patients with a kalemia<3.5mmol/L was 8% (n = 101). Renal insufficiency (OR [95% CI] = 3.56[1.94–6.52], p-value <0.001) and hemoglobin <12g/dl (OR [95% CI] = 2.62[1.50–4.60], p-value = 0.001) were associated with hyperkalemia. Female sex (OR [95% CI] = 1.31[1.03–1.66], p-value = 0.029), age <45years (OR [95% CI] = 1.69 [1.20–2.37], p-value = 0.002) and the use of thiazide diuretics (OR [95% CI] = 2.04 [1.28–3.32], p-value = 0.003), were associated with hypokalemia<4mmol/l. Two patients died in the ED and 629 (52.7%) were hospitalized. Hypokalemia <3.5mmol/L was independently associated with increased odds of hospitalization or death (OR [95% CI] = 1.47 [1.00–2.15], p-value = 0.048).

## Conclusions

Hypokalemia is frequently found in the ED and was associated with worse outcomes in a low-risk ED population.

## Introduction

Potassium disturbances are common and have been associated with increased mortality in several populations, namely in those with diabetes [1–3], chronic kidney disease (CKD) [2–7], myocardial infarction (MI) [8, 9], hypertension [9, 10] and heart failure (HF) [2, 3, 6, 9, 11–14]. In these populations, potassium levels have been associated with outcomes in a U-shaped manner, where both hypo- and hyperkalemia portend worse prognosis.

Most studies in the ED have been performed in specific populations, including CKD [15], acute MI [8, 16] and HF [12] or subsequently admitted to critical care [17], or not centered on potassium levels [18, 19] or otherwise descriptive of severe hyperkalemia [20–23]. Among these, only one study was conducted prospectively [12]. Patients included in these previous studies thus represent a high-risk subset, and not the general population presenting at the ED except for one retrospective and monocentric study [21]. On the other hand, potassium disturbances and its clinical implications have been less described among people from the general population presenting at the emergency department (ED [5, 6, 8, 17, 18, 24, 25].

The present study aims to assess the prevalence of dyskalemia in the general population presenting at the ED without unstable medical conditions. Moreover, we aim to describe the factors and the prognostic implications of potassium disturbances.

## Materials and methods

The present study is a post-hoc analysis of the Adverse Drug Events and Self-medication in Emergency Departments (ADES-ED) cohort [26], which aimed at determining the frequency and severity of adverse drug reactions (ADR) related to self-medication (ADR-SM) among patients attending the ED, and also to describe their main characteristics.

The ADES-ED study was a prospective, cross-sectional, observational study conducted over a period of 8 weeks (1 March to 20 April 2010), in the ED of 11 French academic hospitals. The centers were: CHU of Clermont-Ferrand, CHU of Caen, CHU of Toulouse, CHU of Nantes, CHU of Paris-Hôtel-Dieu Hospital, CHU of Rennes, CHU of Paris-Saint Antoine Hospital, CHU of Paris-Mondor Hospital, CHU of Grenoble, CHU of Paris-Cochin Hospital, and CHU of Angers.

from Baxter and Fresenius, research support from Sphingotec, and consulting fees from Novartis; LAV, JPF, NA, BTP, EM, JMB: nothing to disclose. No author has any specific patents or products related to this manuscript to declare. The information presented within the Competing Interests statement does not alter our adherence to PLOS ONE policies on data or materials sharing.

## Recruitment method in the ADES-ED cohort

The source population was all subjects 18 years of age and older who were likely to be admitted to a participating University Hospital emergency department during the study period.

The study population was identified by a pair of students (hospital medical students and fifth year hospital-university pharmacy students) trained in patient selection and data collection, with the help of the Nurse Organizer of the Reception Centre.

A high volume of visits in participating EDs precluded an uninterrupted prospective screening for inclusion throughout the study period. Therefore, we included patients presenting during different time-periods that were randomly selected (at each center level) by a central computer, thereby limiting selection bias associated with ED circadian cycles [26].

Criteria for inclusion:

All adult patients 18 years of age and older admitted to one of the 11 investigative centers during one of the collection periods.

Exclusion criteria:

- deliberate drug intoxication;

- patient refusal to participate in the study;

- patient's inability to give consent.

## ADES-ED study design

Patient demographics were collected from administrative data of each participating center over the same 8-week enrolment period. These data were compared with the overall study population to verify the representativeness of the studied ED population [26]. All adult patients presenting to the ED during one of the randomly assigned time slots were eligible for study participation.

All patients entering the study signed an informed consent form.

Self-medication and medical history were assessed by a standardized questionnaire. This questionnaire was developed and validated by Roulet et al. [27]. The questionnaire contained a set of 20 closed-ended questions exploring all indications and dimensions of self-medication, and data regarding medical history.

## Post-hoc study

Only patients with available potassium measurements were included in the present study.

Hypokalemia was defined according to two independently explored thresholds ($<4$ and $<3.5$mmol/L) [10], while hyperkalemia was defined as a plasma potassium $>5$mmol/L [28]. We chose to use two thresholds because most authors define a potassium level $<4$mmol/L as hypokalemia [29], but potassium levels $<3.5$mmol/L may have a stronger prognostic association[3] and were considered by others [30]. A normal potassium range was considered when potassium levels were between 4 and 5mmol/L or 3.5 to 5mmol/L depending on the threshold considered. CKD was defined as eGFR $<60$ ml/min/1.73 m$^2$. The primary study endpoint was all-cause hospitalization or death.

The study protocol and patient informed consent procedures were approved by the Ethics Committee (St. Etienne CHU on February 10 2008, ref:N/A) and by the Committee on Information in Health Research (CCTIRS ref:08.369/CNIL ref:AT/YPA/SV/SN/GDP/EM/AR081393) according to French regulations in clinical research.

**Table 1. Characteristics of the study population overall and by potassium level categories (mmol/L).**

| | Overall | By potassium category | | | |
|---|---|---|---|---|---|
| *Potassium concentrations (mmol/L)* | | <4.0 | 4.0–5.0 | >5.0 | |
| N | **1242** | **620** | **549** | **73** | |
| *Variables* | | | | | **p-value** |
| Age (years) | 57.2 ± 22.3 | 53.2±22.0 | 59.5±22.1 | 72.3±16.1 | <0.001 |
| Female gender | 634 | 342 (55.2%) | 263 (47.9%) | 29 (39.7%) | 0.006 |
| SBP (mmHg) | 139.2 ± 25.4 | 138±25 | 140±24 | 140±32 | 0.30 |
| DBP (mmHg) | 77.7 ± 15.3 | 78±15 | 77±14 | 72±16 | 0.011 |
| Heart rate (bpm) | 85.7 ± 19.4 | 85±18 | 86±19 | 86±21 | 0.72 |
| Diabetes | 132 (10.6%) | 7 (1.1%) | 10 (1.8%) | 2 (2.7%) | 0.43 |
| Hypertension | 436 (35.1%) | 189 (30.5%) | 202 (36.8%) | 45 (61.6%) | <0.001 |
| Heart failure | 120 (9.7%) | 8 (1.3%) | 10 (1.8%) | 2 (2.7%) | 0.56 |
| Atrial fibrillation | 90 (7.9%) | 7 (1.1%) | 9 (1.6%) | 0 (0.0%) | 0.45 |
| Alcoolism | 42 (3.7%) | 45 (7.9%) | 38 (7.5%) | 7 (11.1%) | 0.60 |
| eGFR (ml/min/1.73m$^2$) | 92.4 ± 56.6 | 90±28 | 82±30 | 53±33 | <0.001 |
| eGFR <60 ml/min/1.73m$^2$ | 223 (18.0%) | 53 (10.6%) | 116 (18.1%) | 42 (59.2%) | <0.001 |
| Glucose (mmol/L) | 6.4 ± 2.4 | 6.2±2.3 | 6.5±2.5 | 6.6±2.3 | 0.18 |
| Na$^+$ (mmol/L) | 138.9 ± 4.2 | 139±3 | 138±4 | 137±6 | 0.002 |
| K$^+$ (mmol/L) | 4.1 ± 0.6 | 3.7±0.2 | 4.3±0.2 | 5.5±0.6 | <0.001 |
| Hemoglobin (g/dL) | 13.3 ± 2.2 | 13±1 | 13±2 | 11±2 | <0.001 |
| NSAIDs | 168 (13.5%) | 87 (14.0%) | 75 (13.7%) | 6 (8.2%) | 0.39 |
| MRAs | 41 (3.3%) | 12 (1.9%) | 21 (3.8%) | 8 (11.0%) | <0.001 |
| ACEi/ARB | 312 (25.1%) | 122 (19.7%) | 152 (27.7%) | 38 (52.1%) | <0.001 |
| Beta-blockers | 217 (17.5%) | 83 (13.4%) | 112 (20.4%) | 22 (30.1%) | <0.001 |
| Corticosteroids | 76 (6.1%) | 30 (4.8%) | 36 (6.6%) | 10 (13.7%) | 0.010 |
| CCBs | 143 (11.5%) | 67 (10.8%) | 59 (10.7%) | 17 (23.3%) | 0.005 |
| Insulin | 73 (5.9%) | 28 (4.5%) | 33 (6.0%) | 12 (16.4%) | <0.001 |
| Loop-diuretic | 155 (12.5%) | 58 (9.4%) | 79 (14.4%) | 18 (24.7%) | <0.001 |
| Thiazide-diuretic | 115 (9.3%) | 62 (10.0%) | 42 (7.7%) | 11 (15.1%) | 0.081 |
| K binders | 11 (0.9%) | 3 (0.5%) | 5 (0.9%) | 3 (4.1%) | 0.007 |
| K supplements | 67 (5.4%) | 32 (5.2%) | 26 (4.7%) | 9 (12.3%) | 0.025 |
| Deaths at emergency | 2 (0.2%) | 0 (0.0%) | 2 (0.4%) | 0 (0.0%) | 0.28 |
| Hospitalization in ward | 629 (52.7%) | 300 (49.8%) | 288 (55.1%) | 41 (59.4%) | 0.11 |
| Hospital critical care | 16 (1.3%) | 5 (0.8%) | 9 (1.7%) | 2 (2.9%) | 0.22 |

SBP, systolic blood pressure; DBP, diastolic blood pressure; RR, respiratory rate; ACEi/ARBs, angiotensin converting enzyme inhibitors/angiotensin receptor blockers; NSAIDs, nonsteroidal anti-inflammatory drugs; eGFR, estimated glomerular filtration rate (CKD-EPI formula), MRAs, mineralocorticoid receptor antagonists; CCBs, Calcium channel blockers.

## Statistical analysis

Continuous variables are expressed as mean ± standard deviation (SD) if normally distributed or as median and interquartile range if the distribution was skewed. Categorical variables are expressed as frequencies and proportions (%). Association between patient characteristics and dyskalemia was assessed using multinomial logistic regression including all variables with a p value < 0.1 in Table 1 followed by a stepwise backward procedure for retaining the variables selected at a p value < 0.05. Associations with the in-ED combined outcome (hospitalization and/or death) were studied with logistic regression models, using potassium as independent variable and adjusting for age, gender, estimated glomerular filtration rate, hemoglobin and

thiazide diuretics, found to be associated with potassium levels in the previous model. We adjusted on age, gender, estimated glomerular filtration rate, hemoglobin and Thiazide: patients with K>5 were indeed older, had lower eGFR and lower hemoglobin (Table 1), all of which being associated with higher risk for death.

Continuous variables were categorized to attain log-linearity. Odds-ratios are presented with their 95% confidence intervals as OR (CI 95%).

All analyses were performed using $R^{\circledR}$ software.

## Results and discussion

### Characteristics of the study population

Overall, 3017 patients were included in the main study, while 1242 patients with available potassium measurements were included in the present sub-study (Fig 1). The chief complaint was abdominal pain in approximately one-quarter of the patients, with the other most frequent complaints being cardiovascular diseases, trauma and weakness.

Patient characteristics are presented in Table 1. The mean age was 57.2±22.3 years. Approximately one third of the study population presented with a history of hypertension, 11% had diabetes mellitus, and 10% had a history of HF. The mean estimated glomerular filtration rate (eGFR) was 92±57 ml/min/1.73m$^2$ (n = 223, 18% had an eGFR <60 ml/min/1.73 m$^2$). A minority of patients had potassium levels above 5mmol/L (n = 73; 6%). Of note, very few patients had a potassium level above 5.5mmol/L (n = 20; 2%), or above 6mmol/L (n = 1; 0.1%).

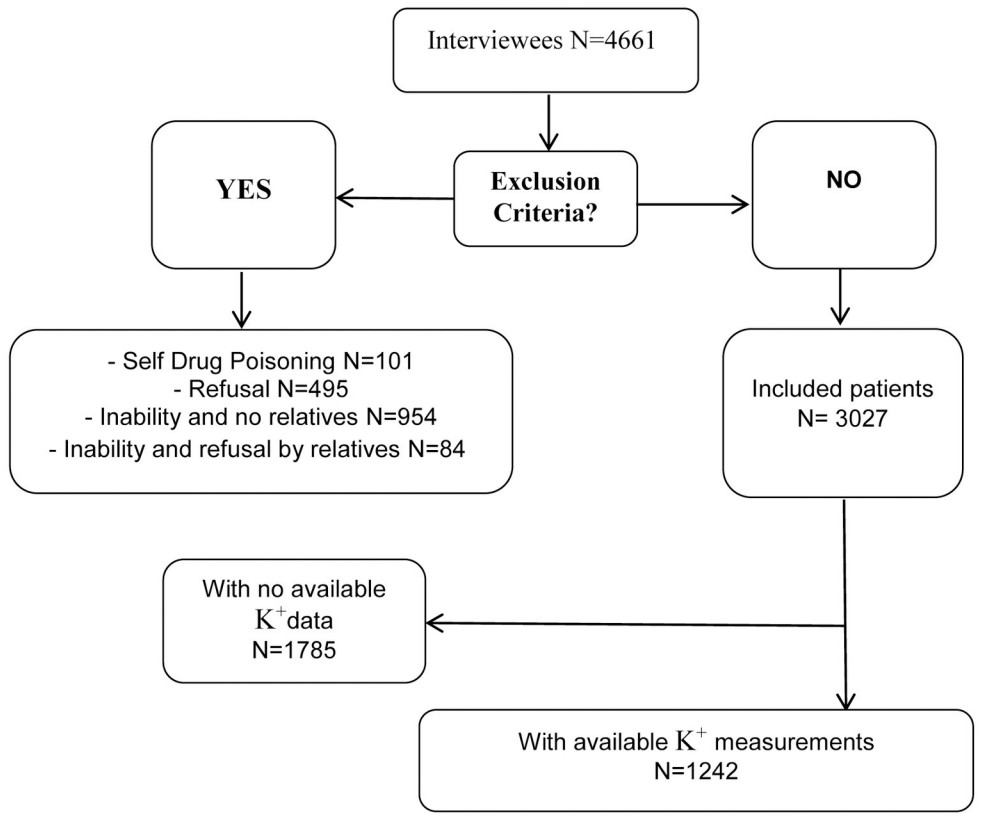

**Fig 1. Flowchart.**

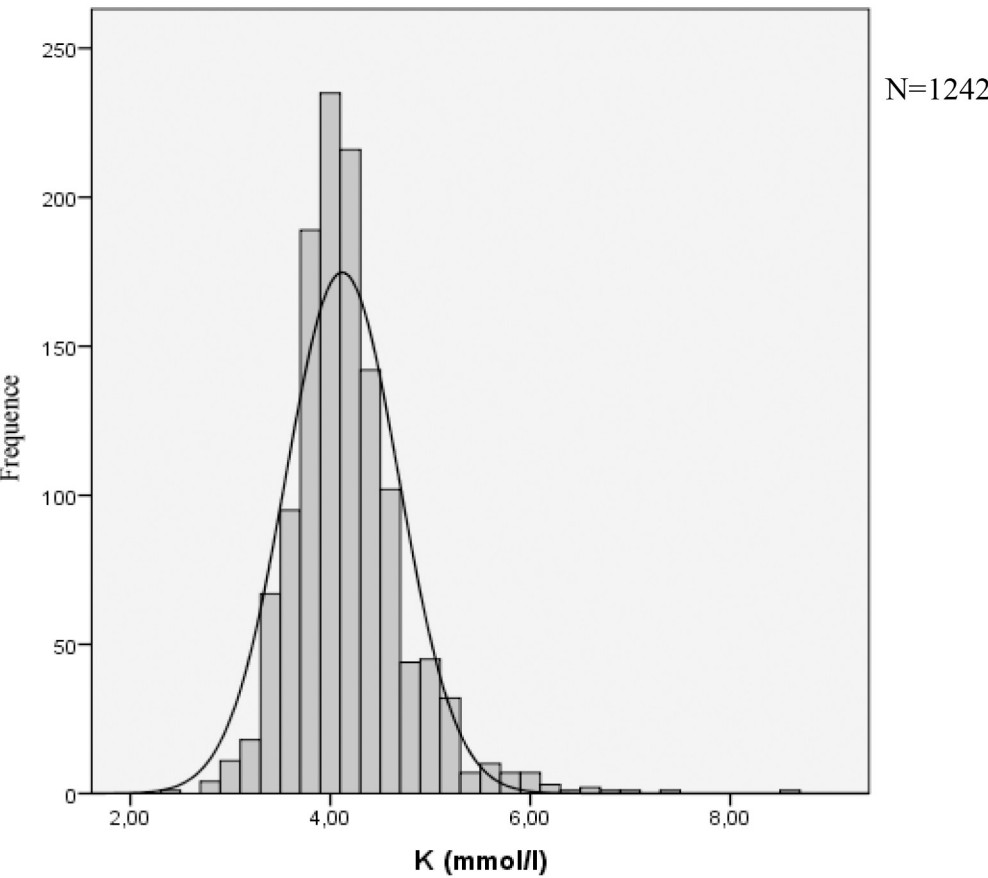

**Fig 2. Potassium distribution.**

In contrast, hypokalemia below 4mmol/L was highly prevalent (n = 620; 50%), while fewer patients displayed hypokalemia below 3.5mmol/L (n = 101; 8%).

Potassium distribution is presented in Fig 2. Patients with hypokalemia were younger and more often female, whereas patients with hyperkalemia had a lower eGFR and hemoglobin (compared with patients with normokalemia).

Patients treated with angiotensin-converting enzyme (ACEi) and angiotensin receptor blockers (ARB), mineralocorticoid receptor antagonists (MRA), calcium channel blockers (CCB), loop diuretics, potassium supplements and insulin had a higher hyperkalemia rate (Table 1).

### Factors associated with hypokalemia

In multivariate analyses, female gender (OR [95%CI] = 1.31 [1.03–1.66], p-value = 0.029), younger age (age <45 years: OR [95% CI] = 1.69 [1.20–2.37], p-value = 0.002) and thiazide diuretic use (OR [95% CI] = 2.04 [1.28–3.32], p-value = 0.003) were associated with hypokalemia <4mmol/L (Table 2).

In a further sensitivity analysis of hypokalemia <3.5mmol/L (with 3.5-5mmol/L as a reference), only thiazide diuretics were found associated with hypokalemia (OR [95% CI] = 2.32 [1.28–4.17], p-value = 0.005) (S1 Table).

**Table 2. Multinomial logistic regression analysis targeting factors associated with dyskalemia.**

| Variable | Hypokalemia OR (95% CI) | p-value | Normokalemia (Reference) | Hyperkalemia OR (95% CI) | p-value |
|---|---|---|---|---|---|
| Female gender | 1.31 (1.03–1.66) | 0.029 | - | 0.54 (0.31–0.92) | 0.024 |
| Age <45 year | 1.69 (1.20–2.37) | 0.002 | - | - | - |
| eGFR <60 ml/min/1.73m$^2$ | 0.64 (0.44–0.93) | 0.020 | - | 3.56 (1.94–6.52) | <0.001 |
| Thiazide diuretics | 2.04 (1.28–3.32) | 0.003 | - | - | - |
| Hb <12 g/dL | - | - | - | 2.62 (1.50–4.60) | 0.001 |

eGFR, estimated glomerular filtration rate based on the CKD-EPI formula; Hb, hemoglobin.

All variables with a p-value < 0.1 in Table 1 were introduced into the model. A backward selection process was conducted with a 500x sampling bootstrap method.

Variables were categorized to attain log-linearity.

Hypokalemia defined as K$^+$ <4.0mmol/L.

Normokalemia defined as K$^+$ 4.0–5.0mmol/L.

Hyperkalemia defined as K$^+$ >5.0mmol/L.

## Factors associated with hyperkalemia

Renal insufficiency (OR [95% CI] = 3.56 [1.94–6.52], p-value <0.001) and hemoglobin <12g/dl (OR [95% CI] = 2.62 [1.50–4.60], p-value = 0.001) were found associated with hyperkalemia, while female gender was negatively associated (OR [95% CI] = 0.54 [0.31–0.92], p-value = 0.0024) (Table 2).

## Hospitalization and death

Two patients died in the ED, 629 (52.7%) were hospitalized and 16 (1.3%) were admitted to an intensive care unit (Table 3). Fig 3 presents the relationship between potassium level in blood and the combined outcome of hospitalization or death, the latter of which was found to be U-shaped, with a nadir at approximately 4.2mmol/L. In multivariate analysis, potassium was not associated with the combined outcome (Table 3).

In a sensitivity analysis, hypokalemia< 3.5mmol/L was associated with the combined outcome of hospitalization or death (OR [95% CI] = 1.47 [1.00–2.15], p-value = 0.048) (Table 4).

Our study reports the prevalence, associated factors and prognostic implications of dyskalemia assessed in patients presenting at the ED in the absence of unstable medical illness. Hypokalemia was present in almost 50% of these patients, particularly among those less than 45 years old, in women, and those taking thiazide diuretics. Importantly, hypokalemia below 3.5mmol/L was associated with poorer outcomes. Conversely, hyperkalemia was less frequent in this population and not associated with adverse prognosis.

Our population characteristics are comparable to other low-risk cohorts [31, 32]. Moreover, similarly to Krokager et al. [33], who reported short-term mortality risk of potassium levels in

**Table 3. Associations between K$^+$ levels and the outcome of hospitalization or death.**

| K$^+$ levels (mmol/L) | Crude OR (95% CI) | p-value | Adjusted OR (95% CI)* | p-value |
|---|---|---|---|---|
| K$^+$ <4.0 | 0.75 (0.59–0.95) | 0.015 | 0.89 (0.69–1.15) | 0.38 |
| K$^+$ 4.0–5.0 (ref.)** | - | - | - | - |
| K$^+$ >5.0 | 1.25 (0.75–2.07) | 0.40 | 0.68 (0.39–1.17) | 0.16 |

*Model adjusted for age, gender, estimated glomerular filtration rate, hemoglobin and thiazide diuretics.

**The reference group is normokalemia: 4.0–5.0mmol/L.

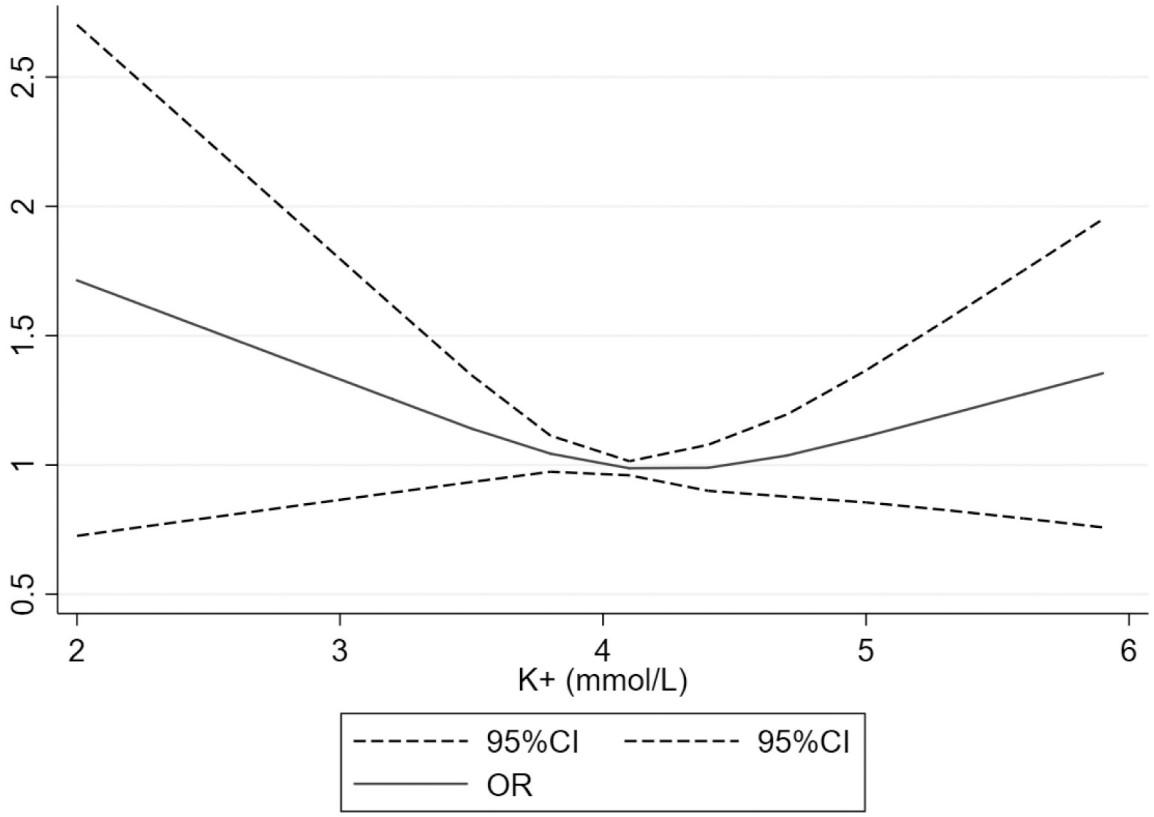

**Fig 3. Relationship between potassium level in blood and combined outcome of hospitalization or death.** *Model adjusted for age, gender, estimated glomerular filtration rate, hemoglobin and thiazide diuretics.

hypertensive patients from the nationwide Danish registry, or in the Lindner et al. ED retrospective monocentric study [21], women were globally more susceptible to hypokalemia. The hypokalemia rates found in the present study were higher than in other reports [20, 21, 34–36].

For external validity purposes, we compared our prevalence results with another ED French cohort—the PARADISE cohort "Pathway of dyspneic patients in Emergency", consisting of including all patients over one year aged 18 years or older admitted to the ED of the Nancy University Hospital (France) for dyspnea., as previously reported by our group [37].

In this setting, encompassing various nosological conditions, among 1318 dyspneic patients with potassium measurements, hypokalemia below 4mmol/ L was present in 36.6% of the

**Table 4. Associations between K⁺ levels and the outcome of hospitalization or death.**

| K⁺ levels (mmol/L) | Crude OR (95% CI) | p-value | Adjusted OR (95% CI)* | p-value |
|---|---|---|---|---|
| K⁺ <3.5 | 1.37 (0.96–1.96) | 0.079 | 1.47 (1.00–2.15) | 0.048 |
| K⁺ 3.5–5.0 (ref.)** | - | - | - | - |
| K⁺ >5.0 | 1.51 (0.92–2.47) | 0.10 | 0.61 (0.35–1.08) | 0.09 |

*Model adjusted for age, gender, estimated glomerular filtration rate, hemoglobin and thiazide diuretics.

**The reference group is normokalemia: 3.5–5.0mmol/L.

patients, which was similar to what we found herein. Hyperkalemia above 5mmol/L was present in 6.8% of the patients.

An increase in the in-hospital short-term morbidity-mortality was observed in patients with potassium levels <3.5mmol/L, a finding also observed in another ED study [35] as well as in myocardial infarction patient cohorts [8] and in both hypertensive [33] and chronic HF [29] patients or in meta-analysis [9]. In specific population, as our results, Zhang et al. [7] demonstrated in meta-analysis that hypokalemia (<3.5mEq/L) was significantly associated with higher mortality risk among patients with CKD and dominant among women. Zhang et al also found that serum K within 3.5–4.0mEq/L among CKD patients was associated with increased all-cause risk. Potassium is an important determinant of myocardial function, and low blood potassium levels may cause arrhythmias and sudden cardiac death by accelerating depolarization, increasing automaticity and lengthening the action potential [29, 38–42].

The use of thiazide diuretics was associated with hypokalemia in our study. Their use may increase urinary potassium loss [43, 44]; hence, an appropriate electrolyte (and creatinine) monitoring is warranted in patients taking thiazides. However, certain studies have shown that the monitoring of potassium levels is suboptimal under potassium-sparing diuretics [45], particularly in primary care patients [46]. Upstream of the ED, i.e. in the primary care setting, there is indeed widespread use of thiazide diuretics, notably as long-run prescriptions in hypertensive patients (53.4% in hypertensive patients in the Danish registry [33], where the administration of diuretics in the low normal potassium level was slightly higher than the administration of ACEIs/ARBs). Thiazide diuretics were also found associated with hypokalemia in the ED in the present study, while "diuretic therapy" was found associated with both hypokalemia and hyperkalemia by Lindner et al. in their retrospective analysis of a single-center database in Switzerland [21]. Thus, an adequate biological monitoring is also warranted in hypertensive patients who are at risk of hypokalemia.

Our study has several limitations. First, the initial reasons for measuring potassium were not recorded and assessed, although the chief complaint of our population was comparable to those of the ADES-ED population (trauma in about one-third of the patients, with the other most frequent complaints being abdominal pain, weakness and cardiovascular diseases) [26], except for trauma which was second. This fact can be easily explained since patients with no potassium measurements were excluded in our study and patients with non-surgical trauma had no blood test. Secondly, data collection on drug consumption did not include dietary supplements, which can also generate iatrogenic effects, especially on potassium levels. Thirdly, the present dataset does not allow ruling out a pseudo-hyperkalemia phenomenon [47], in addition we don't know which machines were used to test their potassium values and what are the hospitals cut-offs for potassium. However, hyperkalemia was very uncommon in the present series. Another important limitation was that mortality was only assessed in the ED, while a more long-term analysis (i.e. during hospital stay) may also be relevant. Lastly, our population only included people who have had a blood test with electrolytes. This fact selects patients who are probably sicker than those who did not have blood tests.

Notwithstanding, the strength of the present study lies in its multicenter and prospective recruitment approach while describing for the first time a population presenting at the ED without unstable medical illness.

## Conclusions

Hypokalemia was frequently found in the ED. It was associated with worse outcomes in a low-risk ED population. Thus, ED patients presenting with hypokalemia should be appropriately treated and monitored both in the ED and after discharge. Furthermore, more effective

prevention strategies encompassing an adequate biological monitoring of hypertensive patients treated with thiazide diuretics should be implemented upstream.

## Supporting information

**S1 Table. Multinomial logistic regression for the factors associated with dyskalemia.** eGFR, estimated glomerular filtration rate based on the CKD-EPI formula; Hb, hemoglobin. All variables with a p-value of less than 0.1 in Table 1 were introduced into the model. A backward selection process was conducted with 500x sampling bootstrap method. Variables were categorized to attain log-linearity. Hypokalemia defined as $K^+$ <3.5 mmol/L (N = 148). Normokalemia defined as $K^+$ 3.5–5.0 mmol/L (N = 1021). Hyperkalemia defined as $K^+$ >5.0 mmol/L (N = 73).
(DOCX)

## Acknowledgments

The authors deeply thank the study steering committee (Françoise Ballereau, Jacques Bouget, Nadine Foucher, Patrice Queneau, Bertrand Renaud, Lucien Roulet, Gerald Kierzek, Aurore Armand-Perroux, Gilles Potel, Jeannot Schmidt and Françoise Carpentier) for granting us access to the clinical database. The authors thank Pierre Pothier for the editing the manuscript.

## Author Contributions

**Conceptualization:** Laure Abensur Vuillaume, João Pedro Ferreira, Patrick Rossignol.

**Data curation:** Laure Abensur Vuillaume.

**Formal analysis:** Laure Abensur Vuillaume, João Pedro Ferreira, Patrick Rossignol.

**Funding acquisition:** Nathalie Asseray, Béatrice Trombert-Paviot.

**Investigation:** Nathalie Asseray, Béatrice Trombert-Paviot.

**Methodology:** Laure Abensur Vuillaume, João Pedro Ferreira.

**Supervision:** João Pedro Ferreira.

**Validation:** João Pedro Ferreira, Patrick Rossignol.

**Writing – original draft:** Laure Abensur Vuillaume, João Pedro Ferreira, Patrick Rossignol.

**Writing – review & editing:** Laure Abensur Vuillaume, João Pedro Ferreira, Nathalie Asseray, Béatrice Trombert-Paviot, Emmanuel Montassier, Matthieu Legrand, Nicolas Girerd, Jean-Marc Boivin, Tahar Chouihed, Patrick Rossignol.

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
