## [Decision Letter · Decision Letter 0]

17 Mar 2020

PONE-D-19-27970

Hypokalemia is frequent and has prognostic implications in stable patients attending the emergency department

PLOS ONE

Dear Professor Rossignol,

Thank you for submitting your manuscript to PLOS ONE. After careful consideration, we feel that it has merit but does not fully meet PLOS ONE’s publication criteria as it currently stands. Therefore, we invite you to submit a revised version of the manuscript that addresses the points raised during the review process.

Two external experts were consulted and both identified some issues to be fixed. Please revise the manuscript according to the reviewers' comments.

We would appreciate receiving your revised manuscript by May 01 2020 11:59PM. To enhance the reproducibility of your results, we recommend that if applicable you deposit your laboratory protocols in protocols.io, where a protocol can be assigned its own identifier (DOI) such that it can be cited independently in the future. For instructions see: http://journals.plos.org/plosone/s/submission-guidelines#loc-laboratory-protocols

We look forward to receiving your revised manuscript.

Kind regards,

Xiongwen Chen, PhD

Academic Editor

PLOS ONE

Journal Requirements:

3. In your Methods section, please provide additional information about the participant recruitment method and the demographic details of your participants.

Please ensure you have provided sufficient details to replicate the analyses such as:

a) a description of how participants were recruited, and b) descriptions of where participants were recruited and the names of the 11 French academic hospitals.

5. Thank you for stating the following in the Financial Disclosure section:

'The National Medicine Academy approved the study design and provided funding. The French Society for Clinical Pharmacy (SFPC) approved the protocol, supported the enrolment and training of pharmacy students and provided funding. The French Association of Pharmaceutical Manufacturers for a responsible self-medication (AFIPA) approved the protocol and the conduct of the study. Sanofi Aventis approved the protocol and the conduct of the study. None of these funding sources intervened in the collection, management, analysis or interpretation of the data; nor in the preparation, review or approval of the manuscript.'

We note that you received funding from commercial sources: AFIPA and Sanofi Aventis

Reviewers' comments:

Reviewer's Responses to Questions

**Comments to the Author**

1. Is the manuscript technically sound, and do the data support the conclusions?

Reviewer #1: No

Reviewer #2: Partly

2. Has the statistical analysis been performed appropriately and rigorously? 

Reviewer #1: No

Reviewer #2: Yes

3. Have the authors made all data underlying the findings in their manuscript fully available?

Reviewer #1: Yes

Reviewer #2: Yes

4. Is the manuscript presented in an intelligible fashion and written in standard English?

Reviewer #1: Yes

Reviewer #2: Yes

5. Review Comments to the Author

Reviewer #1: The following information needs to be included in the manuscript:

1. The inclusion criteria of the ADES-ED study. Without it, the study group is not clearly defined.

2. In line 102, a standardized survey was used. How was the survey validated?

3. Line 116 mentions the PARADISE cohort. A detailed description of the cohort is needed. Why was this group chosen?

4. Lines 106-107: describe why there are two different measures for hypokalemia being used for the study.

5. My previous comment also relates to the two tables (3 & 4) looking at outcomes between a potassium less than 4 and 3.5. It is not clear to me why you need two tables.

6. In the discussion section, you need to address the fact that your population included only those who had blood drawn. This inherently selects for those who are likely sicker than those who did not have blood drawn.

Reviewer #2: This study by Ferreira et al. investigated the prevalence of dyskalemia in ED of multiple medical centers/hospitals across the France. This was done in a prospective and cross-sectional manner. It is interestingly showed that hypokalemia occurs most often in ED patients while hyperkalemia is rare. Though hypokalemia sounds like “low-risk” but is associated worse outcomes in terms of hospitalization and death. While this is very interesting, there are a couple of issues to be addressed:

1. First of all, according to Table 1, there is more frequent hospitalization (including in ward and critical care) of patients with hyperkalemia than of patients with hypokalemia or normokalemia. This agrees with more treatments with cardiovascular drugs such as ACEi, CCB, and diuretics, suggesting these patients had CVD more frequently (e.g., hypertension). Please explain this is contradicting with Table 4. In Table 4, the crude OR was 1.51 for the association between hyperkalemia and outcomes but adjusted OR showed an OR<1.0 (0.61). Why is that and what does this mean?

2. Why in Table 3 and Table 4, K+>5.0 had different OR and p values for the outcome of hospitalization or death?

3. The comparison between your study with the French cohort of 1318 patients should be removed from the abstract and the results part. It is appropriate to keep that in the discussion.

4. What does MRAs stand for in Table 1? Please make sure you have spelled out all abbreviations in the text and notes for tables.

6. PLOS authors have the option to publish the peer review history of their article (what does this mean?). If published, this will include your full peer review and any attached files.

Reviewer #1: No

Reviewer #2: No

---

## [Author Response · Author response to Decision Letter 0]

14 May 2020

Hypokalemia is frequent and has prognostic implications in stable patients attending the emergency department

Answers to reviewers

Journal Requirements

http://www.journals.plos.org/plosone/s/file?id=wjVg/PLOSOne_formatting_sample_main_body.pdf

Response:

Done

and http://www.journals.plos.org/plosone/s/file?id=ba62/PLOSOne_formatting_sample_title_authors_affiliations.pdf

Response:

Done

Response: 

Thank you. The questionnaire has been previously published.

A copyright nevertheless prevents us from publishing it in our article. but we have now added the citation and details about this questionnaire Lines 120-121 in the version with tracked changes and Lines 101-102 in the new version:

“This questionnaire was developed and validated by Roulet et al in 2008.”

3. In your Methods section, please provide additional information about the participant recruitment method and the demographic details of your participants.

Please ensure you have provided sufficient details to replicate the analyses such as:

a) a description of how participants were recruited, and b) descriptions of where participants were recruited and the names of the 11 French academic hospitals.

Response:

Thank you for that remark. The ADES-ED Cohort population was recruited from the emergency departments of 11 French hospitals. The participating centers were: CHU of Clermont-Ferrand, CHU of Caen, CHU of Toulouse, CHU of Nantes, CHU of Paris-Hôtel-Dieu Hospital, CHU of Rennes, CHU of Paris-Saint Antoine Hospital, CHU of Paris-Mondor Hospital, CHU of Grenoble, CHU of Paris-Cochin Hospital, and CHU of Angers (lines 87-90 in the version with tracked changes and lines 72-75 in the new version). 

All subjects 18 years of age and older who were likely to be admitted to a UHC emergency department during the study period, were recruited. The exclusion criteria were: deliberate drug intoxication; patient refusal to participate in the study; and patient's inability to give consent. 

The study population was identified by a pair of students. We have added a section dedicated of recruitment method to the lines 91-96 in the version with tracked changes and lines77-86 in the new version: 

“Recruitment method in ADES-ED cohort

The source population was all subjects 18 years of age and older who were likely to be admitted to a participating University Hospital emergency department during the study period. 

The study population was identified by a pair of students (hospital medical students and fifth year hospital-university pharmacy students) trained in patient selection and data collection, with the help of the Nurse Organizer of the Reception Centre.

And lines 101-107 in the version with tracked changes and line 87-93 in the new version: 

“Criteria for inclusion: 

All adult patients 18 years of age and older admitted to one of the 11 investigative centers during one of the collection periods. 

Exclusion criteria: 

- deliberate drug intoxication; 

- patient refusal to participate in the study; 

- patient's inability to give consent. “ 

Response: 

The dataset we used was a subset of the main ADES-ED study. Since there are ethical restrictions for data access per French Regulation (sensitive healthcare data), we got access to it via Mrs Trombert, acting as a deputy of the trial sponsor and of the ADES-ED trial steering committee. More broadly, requests for access require the agreement of the study investigators and the study sponsor studying the applicant's project. You can contact this committee by mail: Ms Béatrice Trombert-Paviot. (Beatrice.TROMBERT-PAVIOT@chu-st-etienne.fr ; CHU of Saint Etienne, France) or Pr Patrice Queneau (pat-queneau@orange.fr, Saint Etienne university, France). 

5. Thank you for stating the following in the Financial Disclosure section:

'The National Medicine Academy approved the study design and provided funding. The French Society for Clinical Pharmacy (SFPC) approved the protocol, supported the enrolment and training of pharmacy students and provided funding. The French Association of Pharmaceutical Manufacturers for a responsible self-medication (AFIPA) approved the protocol and the conduct of the study. Sanofi Aventis approved the protocol and the conduct of the study. None of these funding sources intervened in the collection, management, analysis or interpretation of the data; nor in the preparation, review or approval of the manuscript.'

We note that you received funding from commercial sources: AFIPA and Sanofi Aventis

Response: 

Our apologies for the confusion; the above statement actually was related to the main study, for which the initial authors also stated that “All authors have declared no potential conflicts of interest regarding this study.” In this post-hoc analysis, there is no funding, as clarified in the cover letter. 

a. Please provide an amended Competing Interests Statement that explicitly states this commercial funder, along with any other relevant declarations<http://www.plosone.org/static/competing.action> relating to employment, consultancy, patents, products in development, marketed products, etc.

Response:

As explained above, there is no funding source for this post-hoc analysis. This is now clarified in the cover letter.

Response:

Done.

Response: 

Dr Rossignol ORCID number is https://orcid.org/0000-0001-8009-3873

Reviewers' comments

Reviewer #1

The following information needs to be included in the manuscript:

1. The inclusion criteria of the ADES-ED study. Without it, the study group is not clearly defined.

Response:

Thanks; we proceeded accordingly and inserted the following text in order to clarify the criteria of the ADES-ED study lines 101-107 in the version with tracked changes and lines 87-93 in the new version: 

“Criteria for inclusion: 

All adult patients 18 years of age and older admitted to one of the 11 investigative centers during one of the collection periods. 

Exclusion criteria: 

- deliberate drug intoxication; 

- patient refusal to participate in the study; 

- patient's inability to give consent. “ 

2. In line 102, a standardized survey was used. How was the survey validated?

Response: 

The questionnaire has been previously published (Roulet et al, reference 27), with a copyright preventing us from publishing it within the present paper. 

This Questionnaire to document self-medication was developed between January and September 2008 by Roulet et al; during the same period, spontaneous report of self-medication use was collected (“reference period”). The questionnaire was tested and reﬁned between October and December 2008 (“test period”). Finally, the QSMB was routinely used between January and December 2009 (“routine period”). After verifying that the patients interviewed during the reference and routine periods were not signiﬁcantly different with regard to baseline characteristics, Roulet et al assessed the utility of questionnaire by comparing the self-medication rates measured during these two periods. 

We have now quoted this paper lines 119-120 in the version with tracked changes and lines101-102 in the new version:

“This questionnaire was developed and validated by Roulet et al (27).”

3. Line 116 mentions the PARADISE cohort. A detailed description of the cohort is needed. Why was this group chosen?

Response:

Thank you for this remark; we now more accurately describe the PARADISE cohort. This cohort "Pathway of dyspneic patients in Emergency”, consisting of including all patients over one year aged 18 years or older admitted to the ED of the Nancy University Hospital (France) for dyspnea. This cohort included a total of 1,589 patients, including 1318 with a potassium measurement.

This cohort was chosen for several reasons: 

- It is a French population of patients presenting in the emergency department.

- Our team made up this cohort. So we had easy access to the data.

We have added the sentence to lines 247-250 in the version with tracked changes and lines 222-225 in the new version: 

“For external validity purposes, we compared our prevalence results with another ED French cohort - the PARADISE cohort "Pathway of dyspneic patients in Emergency”, consisting of including all patients over one year aged 18 years or older admitted to the ED of the Nancy University Hospital (France) for dyspnea., as previously reported by our group (29).”

4. Lines 106-107: describe why there are two different measures for hypokalemia being used for the study.

Response:

Thank you. These cut-offs were chosen to have a better insight on the shape of the associations, as most authors define a potassium level <4mmol/L as hypokalemia (reference 29), but potassium levels <3.5mmol/L may have a stronger prognostic association, and were selected by other groups e.g. recently by Cooper et al Eur J Heart Fail. 2020 (reference 30).

Therefore, we wanted to assess both. 

We have added the sentence to lines 126-129 in the version with tracked changes and lines 109-112 in the new version: 

“We chose to use two thresholds because most authors define a potassium level <4mmol/L as hypokalemia [29], but potassium levels <3.5mmol/L may have a stronger prognostic association and were considered by others [30].”

We now also state that, line 130 in the version with tracked changes and lines 112-113 in the new version that: 

“normal potassium range was considered when potassium levels were between 4 and 5mmol/L or 3.5 to 5mmol/L depending on the threshold considered”

5. My previous comment also relates to the two tables (3 & 4) looking at outcomes between a potassium less than 4 and 3.5. It is not clear to me why you need two tables.

Response: 

We have made two tables since there are actually two different analyses. Indeed, the association of K+<3.5 with outcomes is stronger, therefore, we would like to maintain it for the reader. 

6. In the discussion section, you need to address the fact that your population included only those who had blood drawn. This inherently selects for those who are likely sicker than those who did not have blood drawn.

Response:

Great point, thank you. We have added this sentence in limitation’s section lines 293-294 in the version with tracked changes and lines 264-265 in the new version:

“Lastly, our population only included people who have had a blood test with electrolytes. This fact selects patients who are probably sicker than those who did not have blood tests.” 

Reviewer #2

This study by Ferreira et al. investigated the prevalence of dyskalemia in ED of multiple medical centers/hospitals across the France. This was done in a prospective and cross-sectional manner. It is interestingly showed that hypokalemia occurs most often in ED patients while hyperkalemia is rare. Though hypokalemia sounds like “low-risk” but is associated worse outcomes in terms of hospitalization and death. While this is very interesting, there are a couple of issues to be addressed:

1. First of all, according to Table 1, there is more frequent hospitalization (including in ward and critical care) of patients with hyperkalemia than of patients with hypokalemia or normokalemia. This agrees with more treatments with cardiovascular drugs such as ACEi, CCB, and diuretics, suggesting these patients had CVD more frequently (e.g., hypertension). Please explain this is contradicting with Table 4. In Table 4, the crude OR was 1.51 for the association between hyperkalemia and outcomes but adjusted OR showed an OR<1.0 (0.61). Why is that and what does this mean?

Response:

This point is well taken. In the Table 4, we adjusted on age, gender, estimated glomerular filtration rate, hemoglobin and Thiazide. Indeed, patients with K>5 were older, had lower eGFR and lower hemoglobin, all of which are associated with higher risk for death.

When adjusting on those factors, the increase in the risk of event observed with K>5 is reversed to a rather preventive association (lower than 1). This is because of the major confounding effect of age, eGFR and hemoglobin.

In the revised version of the manuscript we added this sentence to provide this important message to the reader, in the methods section lines 149-151 in the version with tracked changes and lines 129-131 in the new version: 

« We adjusted on age, gender, estimated glomerular filtration rate, hemoglobin and Thiazide: patients with K>5 were indeed older, had lower eGFR and lower hemoglobin (Table 1), all of which being associated with higher risk for death. »

2. Why in Table 3 and Table 4, K+>5.0 had different OR and p values for the outcome of hospitalization or death?

Response:

The association of K>5 with hospitalization/death is changing from one table to the other because of a different reference category. In the Table 4, the reference group (3.5-5) is larger, and at lower risk for event than the reference group (4-5). This directly increases the association of K>5 with outcome in Table 4 vs Table 3. We have now made this clearer in the manuscript by making clear what is the reference category in the methods section (stating now, line 129 in the version with tracked changes and line 112 in the new version that “normal potassium range was considered when potassium levels were between 4 and 5mmol/L or 3.5 to 5mmol/L depending on the threshold considered”) and in the respective tables. 

Please see the revised version of the manuscript. 

3. The comparison between your study with the French cohort of 1318 patients should be removed from the abstract and the results part. It is appropriate to keep that in the discussion.

Response:

We proceeded accordingly and removed this part from the abstract, and these points are detailed in the discussion section. Please see the revised version of the manuscript. 

4. What does MRAs stand for in Table 1? Please make sure you have spelled out all abbreviations in the text and notes for tables.

Response:

Thank you. MRAs stands for mineralocorticoid receptor antagonists. We have checked all abbreviations and added them to the respective legends of the tables. Please see the revised version of the manuscript.

---

## [Decision Letter · Decision Letter 1]

17 Jul 2020

Hypokalemia is frequent and has prognostic implications in stable patients attending the emergency department

PONE-D-19-27970R1

Dear Dr. Rossignol,

We’re pleased to inform you that your manuscript has been judged scientifically suitable for publication and will be formally accepted for publication once it meets all outstanding technical requirements.

Kind regards,

Xiongwen Chen, PhD

Academic Editor

PLOS ONE

Additional Editor Comments (optional):

Reviewers' comments:

Reviewer's Responses to Questions

**Comments to the Author**

1. If the authors have adequately addressed your comments raised in a previous round of review and you feel that this manuscript is now acceptable for publication, you may indicate that here to bypass the “Comments to the Author” section, enter your conflict of interest statement in the “Confidential to Editor” section, and submit your "Accept" recommendation.

Reviewer #2: All comments have been addressed

2. Is the manuscript technically sound, and do the data support the conclusions?

Reviewer #2: Yes

3. Has the statistical analysis been performed appropriately and rigorously? 

Reviewer #2: Yes

4. Have the authors made all data underlying the findings in their manuscript fully available?

Reviewer #2: No

5. Is the manuscript presented in an intelligible fashion and written in standard English?

Reviewer #2: Yes

6. Review Comments to the Author

Reviewer #2: You have adequately addressed my comments. Congratulations! Please make sure that you will make the underlying data for this manuscript fully available to the journal.

7. PLOS authors have the option to publish the peer review history of their article (what does this mean?). If published, this will include your full peer review and any attached files.

Reviewer #2: **Yes: **Xiongwen Chen

---

## [Editor Report · Acceptance letter]

24 Jul 2020

PONE-D-19-27970R1 

Hypokalemia is frequent and has prognostic implications in stable patients attending the emergency department 

Dear Dr. Rossignol:

I'm pleased to inform you that your manuscript has been deemed suitable for publication in PLOS ONE. Congratulations! Your manuscript is now with our production department. 

Kind regards, 

on behalf of

Dr Xiongwen Chen 

Academic Editor

PLOS ONE